# Metachronal Motion of Biological and Artificial Cilia

**DOI:** 10.3390/biomimetics9040198

**Published:** 2024-03-27

**Authors:** Zhiwei Cui, Ye Wang, Jaap M. J. den Toonder

**Affiliations:** 1Department of Mechanical Engineering, Eindhoven University of Technology, P.O. Box 513, 5600 MB Eindhoven, The Netherlands; z.cui@tue.nl (Z.C.); y.wang2@tue.nl (Y.W.); 2Institute for Complex Molecular Systems, Eindhoven University of Technology, P.O. Box 513, 5600 MB Eindhoven, The Netherlands

**Keywords:** metachronal motion, cilia, flow generation, transportation, microrobot

## Abstract

Cilia are slender, hair-like cell protrusions that are present ubiquitously in the natural world. They perform essential functions, such as generating fluid flow, propulsion, and feeding, in organisms ranging from protozoa to the human body. The coordinated beating of cilia, which results in wavelike motions known as metachrony, has fascinated researchers for decades for its role in functions such as flow generation and mucus transport. Inspired by nature, researchers have explored diverse materials for the fabrication of artificial cilia and developed several methods to mimic the metachronal motion observed in their biological counterparts. In this review, we will introduce the different types of metachronal motion generated by both biological and artificial cilia, the latter including pneumatically, photonically, electrically, and magnetically driven artificial cilia. Furthermore, we review the possible applications of metachronal motion by artificial cilia, focusing on flow generation, transport of mucus, particles, and droplets, and microrobotic locomotion. The overall aim of this review is to offer a comprehensive overview of the metachronal motions exhibited by diverse artificial cilia and the corresponding practical implementations. Additionally, we identify the potential future directions within this field. These insights present an exciting opportunity for further advancements in this domain.

## 1. Introduction

Cilia are slender, hair-like microstructures projecting from cells that are present in numerous organisms ranging from unicellular entities to the human body; their typical length ranges from 1 to 30 μm [1,2,3]. The individual cilium exhibits asymmetric beating at low Reynolds numbers, with viscous forces dominating inertial forces due to the small scale of the biological cilia. A single beating cycle of the cilium comprises two phases, namely the effective stroke and the recovery stroke [4,5,6,7,8]. During the effective stroke, the cilium undergoes linear, rod-like movement, whereas in the recovery stroke, it takes a more curved shape, moving often closer to the surface [9,10,11]. A phenomenon often observed in biological cilia is their collective wavelike motion, with adjacent cilia displaying a slight phase lag, which is known as metachronal motion. This coordinated asynchrony can result in various types of wave-like patterns of cilia movement. There are four types of metachronal motion, which can be distinguished by the direction of the wave propagation in relation to the beating motion of individual cilia (see Figure 1). For symplectic metachronal motion, the wave propagation direction is the same as that of the effective stroke; antiplectic metachronal motion is characterized by the wave propagation direction being opposite to the direction of the effective stroke; in dexioplectic metachronal motion, the wave propagates perpendicularly to the effective stroke and the beat is to the right of the wave; finally, laeoplectic metachronal motion involves wave propagation perpendicular to the effective stroke and the beat is to the left of the wave [4,5,8]. In nature, symplectic and antiplectic metachrony are predominant.

Research has indicated that the metachronal motion of the cilia plays an important role in many biological processes, such as mucociliary clearance [1,9,12,13,14,15], fluid transport [16,17,18], and sensory functions [3,19]. The malfunction of the cilia is associated with serious diseases such as cystic fibrosis (CF), primary ciliary dyskinesia (PCD), and chronic obstructive pulmonary disease (COPD) [1]. One of the most common instances of metachronal coordination is observed in the cilia found within the respiratory system of mammals, where the metachronal coordination facilitates the clearance of mucus and the removal of pathogens from the airways [20]. The disorder of the metachronal motion will result in the accumulation of mucus and bacteria in the airways and lead to infections. Furthermore, the metachronal motion of the cilia facilitates the transport of ovum and sperm within the female oviduct [9]. In the case of *Paramecium*, the metachronal motion of the cilia covering its outer surface can propel this microorganism at speeds equivalent to 10 times its own body length per second [17]. Also, numerical studies have shown that the metachronal motion of artificial cilia can enhance flow generation in lab-on-chip devices compared to synchronous cilia motion [17,21]. Inspired by the prominence of metachronal motion in various biological processes, researchers have endeavored to develop artificial cilia that can exhibit metachronal motion.

Research on the metachronal motion of artificial cilia started over a decade ago and is currently undergoing rapid development. Several research groups have developed different methods to fabricate artificial cilia while realizing metachronal motion through various approaches based on the properties of the artificial cilia patch/array. Artificial cilia, as micro-actuators, can be actuated in many ways, including but not limited to electric fields [22,23,24,25], pneumatic pumps [21,26,27], light [28,29,30], and magnetic fields [31,32,33,34,35]. In these systems, the prime goal is mostly to achieve asymmetric motion of the individual artificial cilia, and the metachronal motion must be realized by controlling the phase difference in motion of neighboring cilia. By mimicking the metachronal motion of biological cilia, researchers have been exploring the capabilities of the developed artificial cilia and the added value of metachronal motion in applications such as generating flow in microfluidic chips, transportation of particles, droplets, and mucus, as well as walking robots [3,19,35,36,37,38].

The current state of the topic of metachronal cilia motion presents an exciting stage of development, prompting a comprehensive review of its status and future potential, which is the aim of this paper. We collected all the relevant studies by searching the literature for the keywords “metachronal motion” and “cilia”. Different from existing reviews about cilia, in this review, we focus specifically on the metachronal motion of cilia and its corresponding applications. We start with a brief overview of natural cilia and the metachronal motion shown by them. Then we describe how artificial cilia can mimic biological metachronal motion by reviewing the proposed methods to fabricate and actuate the artificial cilia to achieve metachronal motion for different actuation principles. Subsequently, we review the main applications of metachronal cilia motion that have been studied, including flow generation, particle/droplet transportation, and microrobot locomotion. We conclude the review with a general summary and a forward-looking perspective on the future development of the metachronal motion of artificial cilia.

## 2. Metachronal Motion of Biological Cilia

In this section, we present a comprehensive review of the metachronal motion exhibited by natural biological cilia.

Biological cilia were first observed in 1675 by Antonie van Leeuwenhoek, marking the discovery of these hair-like structures protruding from the surface of cells [39]. In the centuries after this, researchers extensively studied the structure, movement, and functions of the biological cilia in detail. Biological cilia, quite ubiquitous in nature, can be found in many different organisms, tissues, and small creatures. For instance, protozoa, zebrafish, rats, and almost all the tissues of the human body have cilia [11,40,41,42]. Cilia dimensions exhibit considerable variability, ranging from a few micrometers in length to several tens of micrometers, but typically within the range of 1 to 30 μm [3]. Notably, certain cilia, for instance, the flagellum of sperm cells, attain lengths extending into the tens of micrometers [40]. Furthermore, it has been observed that the structures of cilia are different depending on their location, leading to distinct functions. Cilia are responsible for a wide range of physiological functions, including motility, sensing, and signaling [1,43,44,45]. A cilium is composed of two fundamental components: a basal body connecting it to the cell, and an axoneme forming the hairlike protrusion. There are two main categories: motile cilia, with an axoneme characterized by a 9 + 2 microtubular configuration featuring nine peripheral doublet microtubules enveloping two central singlet microtubules, and non-motile or primary cilia, defined by a 9 + 0 microtubular arrangement of the axoneme that lacks the central pair of microtubules [40,41,46,47].

A noteworthy observation in the domain of biological cilia is the prevalence of metachronal motion, a phenomenon of coordinated wavelike movement of arrays of cilia. As explained earlier, there are four basic types of metachronal motion in nature, as illustrated in Figure 1. Morphologically, in symplectic metachrony shown in Figure 1a, the cilia performing the effective stroke are spaced more closely together than the ones performing the recovery stroke, while this is the other way around for antiplectic metachromism in Figure 1b. In dexioplectic and laeoplectic metachronism, shown in Figure 1c,d, respectively, the individual cilia have more freedom of movement than in symplectic and antiplectic metachronism [5,8].

Symplectic metachronal motion has been observed in *Opalina*, a multinucleate protozoon inhabiting the rectal environment of frogs [5,11]. While *Opalina* is not classified as a true ciliate, it possesses numerous ciliary organelles, often referred to as flagella. This organism exhibits a distinctive morphology, taking the form of a slender, flat disk with average dimensions of approximately 200–300 μm in length, 200 μm in width, and a mere 20 μm in thickness [5]. The ciliary organelles are arranged in parallel rows, separated by approximately 3 μm, and the individual cilia within these rows are spaced at intervals of about 0.33 μm. Typically, these cilia measure 10–15 μm in length, although they may extend further towards the posterior end of the organism [5]. Figure 2a(i) illustrates that these rows generate metachronal waves; the individual cilia show a beat pattern that can be represented approximately by Figure 2a(ii), especially when disregarding minor lateral movements in their three-dimensional beats. The cilia lie close together throughout their beat, resulting in a continuous wave outline, as shown in Figure 2a(iii) [5].

The antiplectic metachronal motion has been observed in the marine organism Pleurobrachia, which possesses eight rows of comb plates, as schematically represented in Figure 2b(i) [5]. A typical comb plate is 800 μm long and has a base measuring 30 μm by 600 μm. These plates are aligned in parallel planes and are arranged at intervals ranging from 300 to 400 μm within their respective rows [5]. Each comb plate comprises approximately 105 cilia, which are interconnected within rows through lamellar connections, providing a degree of cohesion [5]. The whole comb plate functions as a unified entity, and it exhibits a beat pattern when viewed in profile, as illustrated in Figure 2b(ii) [5]. The metachronal coordination is normally antiplectic, as indicated in Figure 2b(i) [5].

Dexioplectic metachronal motion has been found in the *Paramecium*, and the movement of this microorganism has been studied in detail [5]. *Paramecium* measures between 200 and 300 μm in length, possesses cilia approximately 10–12 μm in length, and is organized in longitudinal rows [5]. These rows are separated at a distance of approximately 1.5 μm, while the individual cilia within these rows are spaced approximately 2.5 μm apart [5]. During a complete beating cycle, each individual cilium undergoes a three-dimensional motion, as illustrated in Figure 2c(i) [5]. In positions one to three, the effective stroke, the cilium is more or less straight and undergoes a more rapid and nearly planar phase of movement; in positions four to seven, the recovery stroke, the cilium enters a slower phase in which it has a curved shape and performs a backward motion during which it bends away from the plane of observation [5]. From Figure 2c(ii), it can be seen that the orientation of the effective stroke is towards the right of, and perpendicular to, the direction of propagation of the metachronal wave, so that the metachronism is truly dexioplectic, although the organism as a whole exhibits an antiplectic metachronal pattern at first sight [5]. Interestingly, it was found that the metachronism of the *Paramecium* can be impacted by the viscosity of the surrounding fluid, as shown in Figure 2d [4]. As illustrated in Figure 2d(i), the *Paramecium* exhibits dexioplectic metachrony, a behavior consistent with its typical swimming pattern under normal conditions, characterized by a viscosity of 1 cP. Figure 2d(ii–vi) depict the behavior of *Paramecium* under varying viscosity conditions: 2.6 cP, 5.6 cP, 40 cP, 40 cP, and 135 cP, respectively [4]. It can be seen that there is a clockwise rotation of the power stroke and a transition from dexioplectic metachronism to symplectic metachronism as viscosity increases.

Laeoplectic metachronal motion has been identified in various organisms, including *Chaetopterus, Mollusca, Bryozoa,* and *Ploima*. Two examples are illustrated in Figure 2e [8]. Figure 2e(i) shows the larva of Bugula, while Figure 2e(ii) depicts the cyphonautes larva of a bryozoan [8]. Both figures show the apical view. The cilia are indicated at the circumference, and the arrows indicate the wave motion; the effective stroke of the cilia is directed into the plane of observation.

In conclusion, metachronal cilia motion is a prevalent phenomenon in nature, playing an important role in various organisms. In *Paramecium*, metachronal motion contributes significantly to locomotion, enabling efficient propulsion for swift swimming [11,48]. In mammalian respiratory systems, metachronal motion helps in expelling mucus and clearing debris and pathogens, thereby maintaining respiratory health [2,49]. Additionally, metachronal motion within the reproductive system assists in the transport of the ovum from the fallopian tube to the uterus [50,51]. Overall, metachronal ciliary motion is a remarkable and versatile biological phenomenon that contributes to various important functions in organisms across different ecosystems and life forms. Its coordinated and rhythmic nature is essential for the survival and ecological interactions of many species in the natural world.

## 3. Metachronal Motion of Artificial Cilia

In this section, we will provide an overview of the current state of the art of metachronal motion of artificial cilia based on the existing literature.

Inspired by nature, researchers have tried to mimic the metachronal motion of biological cilia by using artificial cilia, with the intention of applying this technology across diverse scientific domains, for instance, by integrating it into microfluidic chips and other specialized platforms. To date, researchers have developed various fabrication techniques to produce a range of artificial cilia configurations. It is important to clarify that the term “artificial cilia”, as discussed in this article, does not imply an exact replication of the size or structure observed in biological cilia. Artificial cilia can vary in length and need not adhere strictly to a hair-like morphology. Additionally, they can be actuated using diverse stimuli, including pneumatic forces [26,27], light [28,30,51], electric fields [22,23,24], magnetic fields [31,52,53,54], pH variations [55,56], temperature changes [3], and more.

To the best of our knowledge, researchers have successfully realized metachronal motion in artificial cilia through pneumatic actuation [21], light-induced actuation [29], electric field-driven actuation [23], and magnetic field-driven actuation [18,35,36,50,57,58,59]. Consequently, our discussion will focus on these distinct actuation mechanisms and their respective achievements in realizing metachronal motion in artificial cilia.

### 3.1. Metachronal Motion of Pneumatically Driven Artificial Cilia

Pneumatic artificial cilia were first reported by Gorissen and coworkers [26]. These artificial cilia are flexible, hollow cylindrical tubes made of polydimethylsiloxane (PDMS) with an eccentric void, as shown in a cross-section in Figure 3a by the hatched area. If the void were concentric, as in the case of regular tubes, the flexible tube would solely expand in diameter when applying pressure [21,26,37,60]. However, the eccentricity of the void creates an asymmetric stiffness, which causes the cilia to bend when pressurized. Each cilium is connected to a pressure controller, meaning that every single cilium in a cilia array can be actuated independently [21]. The effective stroke of the cilia is defined as the movement from an upright to a bent state when being pressurized. The fabrication process of these cilia is shown in Figure 3b, where we can see that they can be produced by a high aspect ratio molding process. The cavity of the cilia is formed by a polished tungsten carbide microrod with a diameter of 0.61 mm placed in one half of the mold. The other half of the mold consists of a micro-drilled hole that defines the outer dimensions of the cilia (diameter of 1 mm and length of 8 mm). Alignment of both halves of the mold is achieved using locating pins, which are positioned accurately to define the eccentricity (0.14 mm) between the microrod and the drilled hole. After closing the mold, liquid PDMS is poured in and cured, and subsequent demolding finishes the production process. To achieve metachronal motion, the authors fabricated six pneumatic bending cilia positioned in line to form a ciliated surface and applied six fast switching solenoid valves to control the pressure applied to each cilia individually, allowing for imposing pressure waves with an arbitrary phase shift. The authors realized both antiplectic and symplectic metachronal motion by this method, as shown in Figure 3c, in which (i) shows the antiplectic metachronal motion with a phase difference of 45 degrees and (ii) shows the symplectic metachronal motion with a phase difference of 45 degrees as well.

Recently, the same research group enhanced the technique by introducing an additional degree of freedom to the cilia array. This was achieved through the fabrication of a shorter cilium with offsetting hollow structures [21]. Figure 3d illustrates the fabrication process, which closely resembles the first method but with the inclusion of a shorter cilium featuring an offset inner cavity. As these cilia possess a pair of inner cavities rather than a single one, an extra degree of freedom becomes available for the purpose of generating spatial asymmetry, thereby introducing a swept area. This is achieved by applying pressure to each cavity independently, using dedicated pressure sources. The phase difference between the applied trapezoidal pressure profiles serves as a method to finely control the extent of the area swept by the cilia tips, as illustrated in Figure 3e. By using this method, the researchers induced both symplectic and antiplectic metachronal motion as well as synchronous motion with a cilia array having six cilia, as shown in Figure 3f. The superimposed colored lines show the metachronal wave propagation. The continuous line corresponds to the initial metachronal wave formed by connecting the tips of the cilia, while the dashed line represents the wave in the subsequent frame. The effective stroke, determined by the cilia geometry, is to the right; in the case of symplectic metachronal motion, the wave propagates to the right (positive), whereas for antiplectic metachronal motion, it propagates to the left (negative).

### 3.2. Metachronal Motion of Light-Driven Artificial Cilia

Light-driven microactuators offer distinct advantages, including wireless control, scalability, and spatiotemporally selective capabilities. Light-driven artificial cilia are mostly made from liquid crystal polymer networks that are made light-responsive by incorporating photo-responsive elements like azobenzene, which undergo reversible molecular conformational changes upon exposure to UV-range light. These molecular changes are translated to a macroscopic level as the bending of artificial cilia, facilitated by the alignment of liquid crystal molecules within the actuator network.

Palagi et al. [29] reported the generation of wave-like motion using liquid crystal elastomers with incorporated azobenzene, through local time-dependent illumination, causing local expansion or contraction of these materials, as shown in Figure 4a. Complex wavelike motion could be induced by structured light fields generated by an optical system based on a digital micromirror device (DMD) with 1024 × 768 mirrors, as shown in Figure 4b. The researchers realized swimmers could exhibit both symplectic and antiplectic metachronal motions by controlling illumination conditions to manipulate the relative amplitudes of longitudinal and axial deformations, as exemplified in Figure 4c. Figure 4d shows a result, showcasing the displacement of a microswimmer when subjected to light patterns of varying wavelengths (illustrated in green overlays, with corresponding directional indications in green arrows). Although not strictly based on cilia-like structures, this example shows that microswimmers based on metachronal motions, reminiscent of the biological examples in Figure 2, can be realized using light-responsive materials.

### 3.3. Metachronal Motion of Electrically Driven Artificial Cilia

There has been significant research into the development of artificial cilia that are responsive to electrical fields. Den Toonder et al. [22] first reported the successful creation of electrically driven artificial cilia, demonstrating their efficacy in enhancing microfluidic mixing and pumping.

Later, Wang et al. [23] successfully engineered voltage-actuated cilia capable of generating non-reciprocal motion individually and metachronal motion collectively. These cilia have the shape of strips, measuring approximately 50 μm in length, 5 μm in width, and having a thickness of about 10 nm. They consist of a platinum (Pt) thin film with a thickness of 7 nm and are sealed on one side by a passive titanium (Ti) layer anchored to the substrate at one end, as illustrated in Figure 5a. A scanning electron microscope (SEM) image of an assembled artificial cilia array is presented in Figure 5b. The inset in Figure 5b is a scanning transmission electron microscope (STEM) image of a cilium cross-section showing platinum (white) and titanium (black). The artificial cilia are actuated in phosphate-buffered saline (PBS; 1×, pH 7.45) by raising their potential to about 1 V relative to a Ag/AgCl reference electrode, triggering electrochemical oxidation of the exposed Pt surface and resulting in surface expansion and consequent bending of the actuator (Figure 5a, red reaction pathway, left to right). Applying a voltage of approximately −0.2 V reduces the Pt film, which returns the actuator to its initial state (Figure 5a, blue reaction pathway, right to left). Furthermore, the researchers seamlessly integrated the cilia array with complementary CMOS-based microcircuits, enabling untethered control. This integration involved the fabrication of a photovoltaic-powered CMOS clock circuit, as shown in Figure 5c. These integrated components worked together to produce a sequence of phase-shifted voltage signals at a user-defined frequency when exposed to light, as shown in Figure 5d. This approach facilitated the realization of a metachronal wave pattern across the cilia array.

### 3.4. Metachronal Motion of Magnetically Driven Artificial Cilia

Magnetic artificial cilia are the most widely and extensively investigated category within the artificial cilia field, primarily owing to their distinctive advantages, including that the external magnetic field does not require any complex external physical connections, and that the magnetic field does not interfere with (biological) processes within microfluidic chips. Recently, researchers have devised various methodologies to achieve the metachronal motion of magnetic artificial cilia. There are two main approaches to inducing phase differences in a beating magnetic artificial cilia array. One approach is having cilia with different responses to a uniform forcing applied to the entire cilia array, which can be realized by tuning the alignment of magnetic particles within the magnetic artificial cilia to different angles between consecutive cilia, or by varying the geometry of the cilia within an array; the cilia array is then actuated by a rotational uniform magnetic field. The other approach is applying different forcings to each cilium, which is realized by applying a complex external magnetic field to create different magnetic forces and/or torques on neighboring cilia [57]. In this section, we will describe the metachronal motion of magnetic artificial cilia arrays realized by the following three methods: by controlling the magnetic particle distribution in the cilia within the array; by changing the geometry of the cilia within the array; and by controlling the local magnetic field experienced by the cilia within the array.

Controlling the magnetic particle distribution within the artificial cilia.

Researchers have developed different methods to control the magnetic particle distribution of individual artificial cilia within the array. Tsumori et al. [59] reported the realization of the metachronal motion of magnetic artificial cilia by fabricating cilia arrays in which individual cilia have different particle distributions and actuating the cilia array with a rotating permanent magnet, as shown in Figure 6a. The fabrication process of this cilia array involved filling silanized PDMS molds with a composite of iron powder and PDMS in multiple sequential steps. Initially, specific cavities were manually filled, followed by the application of a magnetic flux to induce the formation of aligned iron particle chains and curing the PDMS to fix that alignment. This process was then repeated by filling the remaining cavities while altering the direction of the applied magnetic field, resulting in iron particles aligning in different directions. Lastly, clear PDMS was applied to the top surface of the mold and cured, serving as a substrate for the final fabricated structure. The result was an array of artificial cilia in which the magnetic particle alignment, and hence the magnetization direction, gradually varied across the array. These cilia exhibited distinct responses to the applied actuation magnetic field with metachronal motion, as shown in Figure 6a. A disadvantage of this approach is that the fabrication process is time consuming, and miniaturization is difficult. Gu et al. [58] developed a different method to control the “particle distribution” inside the cilia array. They presented a simple and scalable method to fabricate a stretchable magnetic cilia carpet, which was composed of cilia made of a magnetic composite material, specifically NdFeB particles and Ecoflex, and a non-magnetic stretchable substrate (pure Ecoflex). By stretching the carpet to conform to various three-dimensional geometries, they encoded complex magnetization patterns in the cilia arrays using a magnetizer. The magnetization profile on the cilia carpet will later translate to metachronal wave patterns under a dynamic magnetic field, as shown in Figure 6b. Recently, Zhang et al. [57] successfully controlled particle distribution inside the magnetic artificial cilia in a straightforward way. The metachronal magnetic cilia were fabricated using a micromolding process, during which the distribution of the paramagnetic particles in the cilia was controlled by placing a rod-shaped magnet array, arranged to have an alternating dipole orientation between consecutive magnets, underneath the mold. Because the paramagnetic particles tend to align with the applied magnetic field, neighboring cilia assume different paramagnetic particle alignments, and they will, therefore, have different magnetization directions. Consequently, the geometrically identical cilia exhibit nonidentical bending behaviors in a static uniform magnetic field and perform a metachronal motion in a 2D rotating uniform magnetic field, as shown in Figure 6c.

Controlling the geometry of the artificial cilia.

Another way to realize the metachronal motion of magnetic artificial cilia is by designing the geometry of the individual cilia in the array to be different, which is shown in Figure 6d [50]. Hanasoge et al. [50] fabricated magnetic artificial cilia consisting of micromachined thin magnetic strips of varying length attached at one end to a substrate. They showed that the difference in cilium length controls the phase of the beating motion. Making use of this property, the researchers could realize metachronal waves within a ciliary array with cilia lengths ranging from 60 μm to 600 μm, and actuated by a permanent magnet, as shown in Figure 6d. The phase of a cilium beating is fully defined by the magnetic force acting on the cilium and its elasticity. For a given magnetic flux density, the effect scales with cilium length, enabling the use of cilium geometry to induce metachronal motion.

Controlling the magnetic field.

Metachronal motion of the magnetic artificial cilia can also be realized by controlling the magnetic field applied to each cilium in an array. Zhang et al. [38] achieved antiplectic metachronal motion of a cilia array with identical magnetic artificial cilia by actuating them with a translating magnetic belt having rod-shaped magnets with alternating dipole orientation between consecutive magnets (shown in Figure 6e). The cilia were fabricated with a mixture of PDMS and iron particles using a micromolding method. The mold was fabricated by photolithography. After filling the mixture in the mold, the cilia structures were cured under a perpendicular magnetic field to align the magnetic particles along the cilia length, identically for all cilia. Since the actuation setup generates a non-uniform but periodic magnetic field, the magnetic field applied to each cilium is also different but also time-dependent due to the translation of the belt. This generated a metachronal wave. However, the action system is rather complex, and it is difficult to miniaturize the metachronal motion due to the size constraints imposed by the actuation magnets. Most recently, our research group achieved a breakthrough in miniaturizing the metachronal motion of identical magnetic artificial cilia; see Cui et al. [18]. This advancement was accomplished through the integration of a paramagnetic substructure underneath the cilia array and actuating the array by a simple rotational, uniform magnetic field, as shown in Figure 6f. The underlying principle hinges on the response of the paramagnetic substructure to an external magnetic field, which induces perturbations in the spatial distribution of the field, generating a time-dependent local magnetic field. When appropriately dimensioned, neighboring identical cilia experience distinct magnetic fields at any given moment, thereby giving rise to metachrony in their motion. Notably, we realized both symplectic and antiplectic metachronal motion using this novel approach, as shown in Figure 6f. One of the key advantages of this method is that it enables further miniaturization of artificial cilia metachrony, in contrast to the earlier approaches.

In conclusion, there are four main types of artificial cilia: pneumatically controlled cilia, light-driven cilia, electrically driven cilia, and magnetically actuated cilia. Each method has its advantages and disadvantages.

The pneumatic control of the artificial cilia enables individual cilia triggering and precise control of the phase difference between each cilium. Hence, the nature of the metachrony (symplectic versus antiplectic) can be varied in a straightforward manner. This capability gives the possibility to study the effects of type of metachrony and phase differences on fluid flow generation by artificial cilia, which we will discuss below. The disadvantages of using pneumatic actuation for metachronal cilia motion are the need for many pressure connections and the difficulty of miniaturizing the cilia.

The metachronal motion achieved through light-driven cilia is noteworthy due to its potential for facilitating remote and precise actuation of arrays of artificial cilia. Importantly, this approach often obviates the need for external connections or complex mechanical linkages. The disadvantages are that the response times are often relatively slow (i.e., seconds or more), and light is not a suitable cue for some applications because of the lack of optical access.

Electrically driven artificial cilia can be actuated quickly and offer much flexibility in controlling the timing of actuation through the design of the electrical circuit, which enables metachrony. A disadvantage is that either high voltages are needed [22], or the actuation requires a specific fluidic environment [23]. Conventional electrical actuation requires electrical connections [22], but the solution introduced by Wang et al. [23] makes it possible to remotely control the cilia motion.

Magnetically driven artificial cilia exhibit rapid responsiveness and can penetrate biological tissues smoothly without causing damage. Although the actuation mechanism does not require complex external connections, the setup for actuation itself continues to be somewhat intricate.

## 4. Applications of Metachronal Motion in Artificial Cilia

In this section, we will review the main applications of the metachronal motion created by artificial cilia, which include flow generation, transportation, and microrobot locomotion.

### 4.1. Flow Generation

The generation of fluid flow holds important significance within the field of microfluidic applications. As mentioned before, a main function of biological cilia is to generate fluid flow or propulsion in liquids, in which metachrony often plays an important role, such as in the propulsion of *Paramecium*. Hence, extensive research has been conducted on the fluid flow generation of artificial cilia and on the role of metachronal motion on the induced flows, using both experimental and computational approaches [21,38,61,62,63]. The maximum flow velocities that have been achieved experimentally using metachronal artificial cilia actuation are 19,000 μm/s for pneumatic and 3000 μm/s for magnetic artificial cilia at high Reynolds number conditions (in water), and 500 μm/s for pneumatic and 450 μm/s for magnetic artificial cilia at low Reynolds number conditions (in glycerol) [3]. In this review, we will focus on the proposed mechanisms of flow generation by metachronal cilia motion rather than on comparing quantitative flow speed, which was covered in our previous review [3].

Khaderi et al. [64] studied fluid flow generation by the metachronal motion of symmetrically beating magnetic artificial cilia using numerical simulations. They found the ciliary motion to generate a unidirectional fluid flow in the direction opposite to the metachronal wave. In addition, they found that the flow reaches a maximum for a critical value of the wavelength of the metachronal wave, which depends on the competition between the elastic and viscous forces acting in the ciliary system. The authors explained the observed behavior from two different viewpoints: from a Eulerian point-of-view and a Lagrangian point-of-view, as illustrated in Figure 7a,b; the metachronal wave travels to the right. Figure 7a illustrates the Eulerian viewpoint, showing pressure contours with superimposed streamlines in (i) and the contours of the normalized absolute value of the horizontal component of the velocity in (ii). Due to the instantaneous velocity of the cilia, high-pressure and low-pressure regions develop (the red and blue regions in Figure 7a(i), respectively). Fluid is squeezed out of the high-pressure regions and drawn in by the low-pressure regions, resulting in a series of counter-rotating vortices in the channel. Since the distance between the high-pressure and low-pressure regions opposite to the wave direction is smaller, the pressure gradient is larger, so the counter-clockwise vortices are stronger (see Figure 7a(ii)). As a result, the velocity distribution has a dominant horizontal component to the left, against the metachronal wave. In addition, the authors analyzed the system from a Lagrangian perspective by tracing fluid particles within the fluid domain, as illustrated in Figure 7b. In this figure, the contours represent the normalized absolute velocity in the horizontal direction, while the streamlines indicate the direction of the fluid velocity. The analysis revealed that the out-of-phase motion of cilia generates a net tracer particle velocity towards the left, biggest for the particles closest to the cilia tips, which is opposite to the propagation direction of the metachronal wave.

The analysis of Khaderi et al. [64] showed that metachrony alone can generate net flow, even in the absence of asymmetric motion of the cilia. However, most cilia in nature exhibit non-reciprocal motion. Therefore, researchers also investigated the effect of metachrony of non-reciprocally moving cilia on flow generation, both numerically and experimentally [16,21,63]. These studies found that antiplectic metachronal motion is more effective in enhancing flow generation by synchronous asymmetric cilia motion than symplectic metachronal motion. The common explanation for this effect given in most studies is based on the notion that cilia can have different shielding or obstruction effects on the fluid flow generated by adjacent cilia, depending on the type of metachrony. This can be illustrated by numerical simulation results from Khaderi et al. [63], who reported that antiplectic metachrony leads to a considerable enhancement in flow compared to symplectic metachrony, especially when the cilia spacing is small [62]. Figure 7c depicts the velocity field for both antiplectic (Figure 7c(i), wave traveling to the right) and symplectic (Figure 7c(ii), wave traveling to the left) metachrony at moments of maximum flux during simulations. In both images, the fifth cilium is at the peak of its effective stroke, towards the left. In the case of symplectic metachrony (ii), the flow created by the fifth cilium is obstructed by the close proximity of the fourth cilium, which has just initiated its effective stroke, resulting in the formation of a vortex. Conversely, in the case of antiplectic metachrony (i), the position of the fourth cilium allows unimpeded flow from the fifth cilium, resulting in a greater net fluid flow compared to its symplectic counterpart. Milana et al. [21] reported similar results based on experiments using pneumatically actuated artificial cilia and corresponding numerical simulations, shown in Figure 7d. For small-phase difference metachronal waves, they found that antiplectic coordination increases fluid flow velocity up to 50% compared to a synchronous beating mode, while symplectic metachrony decreases it. Like Khaderi et al. [63], Milana et al. [21] came to the conclusion that these effects are caused by an obstruction mechanism between cilia in the array that hinders effective flow generation in the case of symplectic waves. Also, Dong et al. [35] reported that only antiplectic metachronal waves with specific wave vectors could enhance fluid flows compared with the synchronized case, based both on experiments with magnetic artificial cilia and on simulations. Similar to the explanations given above, the authors proposed that the differences in fluid flow generation for the two metachrony types are due to differences in obstruction effects of fluid flow between adjacent cilia, as illustrated in Figure 7e for antiplectic, synchronized, and symplectic metachrony. Compared to synchronous motion, the cilia in an array with antiplectic metachrony have less blocked local fluid flow from their neighbors during the effective stroke, but more blocked local fluid flow during the recovery stroke; for symplectic metachrony, this is the other way around.

These numerical and experimental studies show that metachrony has an important influence on the fluid generated by arrays of (artificial) cilia, and they give a common explanation for the observed effects. It is important to note that the metachronal motion of the cilia is just one of several influential factors impacting the induced flow rates. Parameters such as cilia geometry, cilia density, actuation frequency, and others can also have significant effects. However, for the sake of brevity and focus, we will not deeply go into this matter here but refer the reader to prior reviews in which these aspects are comprehensively reported [3].

### 4.2. Transportation

Next to generating fluid flow, biological cilia transport matter like mucus, particles (in the airways), and cells (in the fallopian tube). Researchers have attempted to mimic these transportation functions in several studies using metachronal motion-driven artificial cilia for mucus, solid particles, and droplets, which we will review next.

Mucus transportation.

Mucus transportation in the airways by metachronal coordination of natural cilia is essential for healthy respiration [1,15,46,65,66]. To mimic this process in vitro, researchers have carried out studies of mucus transportation by metachronal artificial cilia; the knowledge obtained in these studies could be relevant for understanding impaired mucus clearance and eventually can help in developing treatments.

Pedersoli et al. [66] conducted experiments on the transportation of artificial mucus by the metachronal motion of magnetic artificial cilia, as shown in Figure 8a. The metachronal motion of magnetic artificial cilia was realized by actuating identical cilia arrays using a magnetic belt, as shown in Figure 1e. The researchers studied the transportation properties of both physiological and pathological states of artificial mucus and the impact of the periciliary layer (PCL) on the transport; the PCL is a thin, low-viscosity fluid layer present just below the mucus layer, as indicated in Figure 8a. Figure 8a(ii) shows three situations of mucus transportation. In the first, mucus is in direct contact with the magnetic artificial cilia and with the underlying substrate, and the PCL is absent, as happens in some airway diseases; in this case, no mucus propulsion was observed. In the second and third situations, a PCL layer of different heights is present. The key to mucus transportation turned out to be a balance between the hydrodynamic resistance and the driving force exerted by the cilia.

This first study of mucus transportation by metachronal motion-driven artificial cilia indicates the potential of this approach; however, to reach a model that is physiologically relevant, many more steps need to be taken. In particular, artificial cilia must be miniaturized, and the cilia areal density must be substantially increased to approximate the in vivo situation more faithfully.

Particle transportation.

The controlled and directed transport of particles (both synthetic and biological, e.g., cells) is desirable in both fundamental research and applications such as biomedical and biochemical research, disease diagnostic and therapeutics, drug discovery and delivery systems, and self-cleaning and anti-fouling technologies. Various principles have been developed to achieve controlled particle transportation, especially for microfluidic applications [67]. Recently, metachronal artificial cilia have been studied and investigated to achieve particle transportation.

Inspired by the ciliary structure and the asymmetric wave-form motion observed in the mammalian airway epithelial surface, Ben et al. [68] devised a novel method for transporting polystyrene (PS) microparticles using magnetic artificial cilia. These artificial cilia were designed in both conical and columnar shapes and were actuated using a periodically moving external magnet, as depicted in Figure 8b. Consequently, the cilia array exhibited a periodic wave-like motion. Furthermore, their investigation revealed that the average transport speed of the conical cilia array exceeded that of the columnar arrays. This difference in transport efficiency was attributed to the reduced friction encountered by the conical cilia during the recovery stroke, resulting in a larger resultant force acting on the particles. Additionally, the bending of the conical cilia caused the center of gravity of the PS microsphere to shift forward, promoting transportation due to its inertia. However, the effectiveness of this transportation was also dependent on factors such as the elasticity, height, separation distance, and beating frequency of the cilia array. The highest achieved PS microsphere transportation speed reached approximately 0.09 mm/s at a cycle frequency of around 0.6 Hz for the flexible conical arrays.

Subsequently, within the same research group, Ben et al. [69] successfully demonstrated the underwater transport of silica microspheres using a magnetic artificial cilia array. This array was fabricated and actuated using a periodic external magnetic field as well, as depicted in Figure 8c. Silica particles could be transported by the wave-like motion exhibited by the artificial cilia. The transportation of these particles was found to depend on various factors, including gravity, buoyancy, supporting forces, fluid driving forces, and friction forces acting on the silica sphere. The study further explored the impact of cilia array beating frequency, inter-cilium spacing, cilia array height, microsphere mass, and solution density on transport performance. Notably, the investigation concluded that the peak speed for silica microsphere transportation, amounting to 0.07 mm/s, was attained at a cycling frequency of around 0.6 Hz underwater [69].

The particle transportation capabilities of metachronal artificial cilia have been studied both through numerical simulations and experimental investigations in recent years. Nevertheless, the investigation of the relationship between the metachronal motion of artificial cilia, the particle properties, and particle transportation still deserves further comprehensive exploration.

Droplet transportation.

Droplet manipulation is a rich field in microfluidics, since droplets offer the possibility of manipulating, controlling, and analyzing small volumes of fluids, and droplet microfluidics can facilitate high-throughput experiments. Among many other approaches to transporting droplets, the use of artificial cilia has been studied increasingly in recent years [32,70,71,72,73,74]. Here, we focus on using metachronal cilia motion to achieve droplet transportation.

**Figure 8 biomimetics-09-00198-f008:**
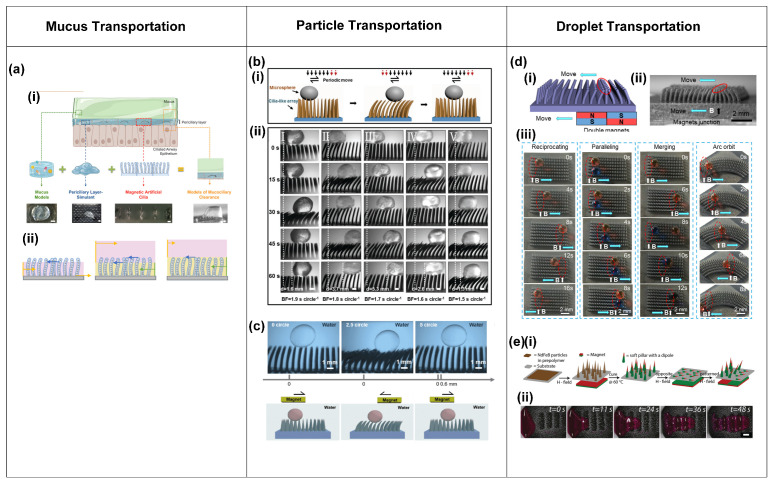
Transportation of mucus, particles, and droplets realized by the metachronal motion of artificial cilia. (**a**) (i) Artificial mucus transportation by metachronal motion of magnetic artificial cilia, illustrated by a schematic (top) and images of transportation results (bottom); (ii) a schematic of mucus transportation by cilia with different thicknesses of periciliary layer (PCL); mucus: pink, PCL: yellow. Reproduced from ref. [66] with permission from Wiley. (**b**) (i) Schematic of microparticle transportation by magnetic artificial cilia that are actuated by a periodic magnetic field; (ii) experimental results of a polystyrene (PS) particle transported by metachronal magnetic cilia at different actuation frequencies. Reproduced from ref. [68] with permission from Wiley. (**c**) Transportation of a silica microparticle in water by metachronal magnetic artificial cilia actuated by a periodic magnetic field. Reproduced from ref. [69] with permission from Science China Press. (**d**) (i) Schematic of the response of an array of magnetic artificial cilia to a moving set of permanent magnets; (ii) microscopy image of the response of an array of magnetic artificial cilia to a moving set of permanent magnets; (iii) droplet transportation by metachronal magnetic cilia. Reproduced from ref. [72] with permission from the American Chemical Society. (**e**) (i) Realization of a dynamic magnetic carpet from a soft silicone matrix with embedded NdFeB particles; (ii) transportation of glycerol droplets by the magnetic cilia carpet. Reproduced from ref. [25] with permission from John Wiley & Sons.

Song et al. [72] demonstrated the successful transportation of droplets through the utilization of unidirectional metachronal waves of magnetic artificial cilia. These waves were dynamically generated through the real-time response of the cilia array to the motion of a set of two permanent magnets with opposite dipoles, thereby enabling the transport of droplets ranging in volume from 1 to 6 μL along a predefined trajectory, as illustrated in Figure 8d. The cilia were made of PDMS containing carbonyl iron powder. After their fabrication, the surface of the cilia was treated with a femtosecond laser to achieve a superhydrophobic, low-adhesion surface. Figure 8d(i,ii) illustrate the metachronal wave made by the cilia in response to the moving magnets. Figure 8d(iii) shows that droplets can be transported along straight or curved lines in the direction of the traveling wave and come to a stop as desired, and that droplets can be merged. Through a thorough analysis of the forces exerted on the droplet, the researchers determined that the droplet moves in the wave direction because the driving force applied to the droplet by the cilia exceeds the frictional forces.

Demirörs et al. [25] achieved the versatile transportation of both fluids and solid materials using the wave-like motion of soft magnetic cilia carpets. The researchers fabricated the magnetic cilia carpet from a soft silicone matrix with embedded neodymium iron boron (NdFeB) particles. By employing varying external magnetic fields, cilia-like soft pillars emerged and exhibited distinct responses, as illustrated in Figure 8e(i). Using this property, the research team successfully demonstrated the transportation of glycerol droplets across the magnetic carpet, as depicted in Figure 8e(ii). The investigation revealed that the glycerol droplet moves in a direction opposite to the motion of the magnetic field wave. The droplet traverses a distance of over 23 mm in less than a minute, showcasing the capabilities of this system [25].

The investigation of droplet transportation by metachronal motion of artificial cilia shows that it has good potential. However, it is important to acknowledge that this research area is still in its early stages, highlighting the need for further in-depth investigations in the future.

Microrobot locomotion.

In the developing field of robotics, scientists and engineers have constantly sought inspiration from the natural world to create machines that mimic the extraordinary capabilities of living creatures. This pursuit has led to the exploration of diverse microscale robotic systems, including those propelled by light-induced actuation, magnetic field manipulation, and other innovative mechanisms [75,76,77,78,79,80,81,82,83]. A noteworthy development in this exploration of biomimicry is the creation of walking robots propelled by metachronal motion, inspired by the locomotion of the millipede, where the legs of the robots are made from cilia-like actuators.

Zhang et al. [57] achieved metachronal motion of magnetic artificial cilia through the precise control of magnetic particle alignment in individual cilia within an array, driven by a uniform rotational magnetic field, as explained above and shown in Figure 6c. The same publication presents a walking microrobot with legs composed of magnetic cilia. The intricate interplay among frictional forces, adhesion forces, and magnetic forces, in combination with the resultant slight bending of the robot body, resulted in forward locomotion of the microrobot, as shown in Figure 9a(i). By simply reverting the rotation direction of the magnetic field, the robot could be made to walk in opposite directions, as shown in Figure 9a(ii). Furthermore, the researchers showcased the remarkable capability of robots to climb across hills, as depicted in Figure 9a(iii). The metachronal motion of the cilia turned out to be crucial for the locomotion: when the cilia moved synchronously, the microrobot did not show any net displacement. Gu et al. [58] also introduced a soft-walking robot with metachronal magnetic artificial cilia as legs, drawing inspiration from the locomotion patterns observed in the African millipede, as illustrated in Figure 9b(i–iii). Their investigation revealed a significant difference in locomotion speed between robots employing antiplectic wave patterns and those with symplectic waves, with the antiplectic wave being much more effective. The authors explained that this difference in speed could be attributed to the opposing curvatures of the substrate during the recovery stroke. As depicted in Figure 9b(iv), the robot body shows an indent at the location of the recovery stroke in the case of symplectic waves, impeding the movement of magnetic cilia due to increased surface friction; conversely, antiplectic soft robots experience bulging of the body at the recovery stroke location, facilitating unconstrained motion of the cilia. This observation sheds light on the mechanism underlying the locomotion of soft-walking robots, offering insights valuable for their design and optimization.

Several experimental studies have explored the capabilities of various soft-walking robots, yet simulation-based research has remained relatively limited. Recently, Jiang et al. [84] established a comprehensive numerical model aimed at investigating the metachronal wave-modulated locomotion of magnetic artificial cilia robots. The authors achieved an accurate replication of the deformation of individual cilia, the coordinated metachronal wave motion exhibited by multiple cilia, and the resulting crawling and rolling locomotion patterns observed in magnetic cilia soft robots. Additionally, they conducted an insightful comparative analysis of substrate deformation in both antiplectic and symplectic metachronal wave soft robots, as shown in Figure 9c. This model holds the potential to provide valuable insights that can guide the design, optimization, and customization of the microrobots.

Next to the magnetic actuation of artificial cilia, light has also been used to create metachronal motion using liquid crystal elastomers with light-responsive azobenzene, as explained before and shown in Figure 4a–d [29,83]. Based on this principle, Palagi et al. designed a microswimmer made from a strip of the light-responsive liquid crystal elastomer (LCE), which is dynamically illuminated with a light pattern, as shown in Figure 9d. Indeed, as shown in the figure, the microrobot exhibits swimming behavior in the direction determined by the anti- or symplectic metachronal wave. Remarkably, these two distinct swimming modes closely resemble the symplectic and antiplectic metachrony types observed in ciliates.

Walking and swimming robots driven by the metachronal motion of artificial cilia hold significant promise for a wide range of applications, including biomedical devices and microsystems. Nevertheless, the realization of their complete capabilities demands a more thorough and comprehensive exploration. Particularly interesting would be to investigate the possibility of creating amphibious microrobots to enhance their versatility and adaptability.

Taken together, metachronal motion offers a broad range of potential applications. While the mechanism of flow generation through metachronal motion has been extensively investigated, the underlying mechanisms linking metachronal motion to other applications, such as transportation and microrobot locomotion, are still in the early stages of understanding.

## 5. Conclusions and Perspectives

Over recent decades, notable advancements have been made in the field of artificial cilia research, including significant progress in both fabrication techniques and actuation strategies. In particular, the last decade has witnessed an increase in the exploration of the metachronal motion of artificial cilia, inspired by the coordinated, wave-like motion of natural cilia found in various organisms. Researchers and engineers have investigated the intricate mechanics governing metachronal motion and have used various actuation methods to achieve metachronal ciliary motion, including pneumatic, optical, electrical, and magnetic techniques. Magnetic artificial cilia have been studied the most since they can be actuated remotely and they provide versatility since metachrony can be realized in diverse ways by either shaping the magnetic actuation field or by controlling individual cilia properties. However, the complicated actuation setup and the shape-isotropic properties of the magnetic artificial cilia still limit the miniaturization of the metachronal motion [3]. Pneumatic artificial cilia offer the unique advantage of completely independent actuation of individual cilia, but their potential is hindered by challenges in miniaturization and the need for intricate connections to actuate the cilia. Overcoming these limitations would significantly broaden their applicability. Light-driven artificial cilia have the advantage of remote actuation of single or clusters of cilia. However, their reliance on a light-transparent environment and their relatively slow response restrict their practical application. Electrically driven artificial cilia face limitations in real-world applications due to their requirement for a high electric field or a specific conductive medium for operation [3]. It will be interesting to combine light and magnetic actuation to realize the metachronal motion of artificial cilia arrays. These two actuation mechanisms do not interfere with each other and may, when combined, provide the possibility to achieve complex shape and motion changes of individual cilia as well as cilia arrays.

In this review, we mainly focus on methods to create the metachronal motion of artificial cilia and the related applications. However, the artificial cilia we have shown in this review still present challenges and limitations. They are still larger than biological cilia, which limits their application to a relatively large scale. Therefore, it is worthwhile to explore the fabrication of artificial cilia arrays with dimensions and aspect ratios close to those of biological cilia. Achieving this goal will involve enhancements in both the materials used and the manufacturing techniques. Specifically, for template-based manufacturing methods, using materials with greater tear strength could significantly improve the demolding success rate for artificial cilia, marking an important step towards more scalable production. To ensure sufficient actuation response, further development towards materials that are both flexible and highly magnetic remains a goal. In terms of creating artificial cilia at the scale of biological ones, it is still a major challenge to increase the areal density of the artificial cilia. Finally, further miniaturizing and integrating the magnetic actuation means will continue to be a necessity for successful application in lab-on-a-chip or organ-on-a-chip systems.

Applications of the metachronal motion of artificial cilia include flow generation, transportation of substances such as mucus, particles, and droplets, and the locomotion of microrobots. Such functions are useful in microfluidic applications, for example, lab-on-a-chip and organ-on-a-chip. Of these functions, fluid flow generation has been investigated the most. Studies have pointed out the relevance of anti-versus symplectic metachrony, with the former leading to more effective fluid transport. The flow velocities that have been achieved using metachronal artificial cilia actuation, for example, 19,000 μm/s for pneumatic and 3000 μm/s for magnetic artificial cilia in water, are well suited to microfluidic applications. The utilization of metachronal motion of artificial cilia for investigating mucus transportation has provided interesting results towards understanding mucociliary clearing of the airways in both healthy and diseased states. However, further development is needed to reach more (patho)physiological relevance, towards miniaturization of the cilia and increasing their spatial density. The application of metachronal motion to particle and droplet transportation not only showcases its capabilities but also suggests promising medical applications, particularly in the domain of drug delivery. However, fundamental research to better understand the underlying transportation mechanisms and facilitate the development of more robust and efficient industrial implementations is necessary. The utilization of metachronal motion in the locomotion of robotic systems remains in its nascent stages of development. While some initial experimental investigations have been conducted, they have primarily focused on applications within controlled environments. Consequently, the broader potential of metachronal motion in versatile terrains, for example, towards amphibious microrobots being capable of both walking in air and swimming in liquid, remains largely unexplored. Moreover, the use of numerical simulations to advance our understanding of mechanisms of robotic locomotion and to enable the directed design of metachronal microrobots also requires efforts in the future.

In conclusion, the study of the metachronal motion of artificial cilia remains a vibrant and evolving research domain. The many emerging developments are catalyzing opportunities for scientific exploration as well as applications, thus shaping a dynamic landscape of innovation.

## Figures and Tables

**Figure 1 biomimetics-09-00198-f001:**
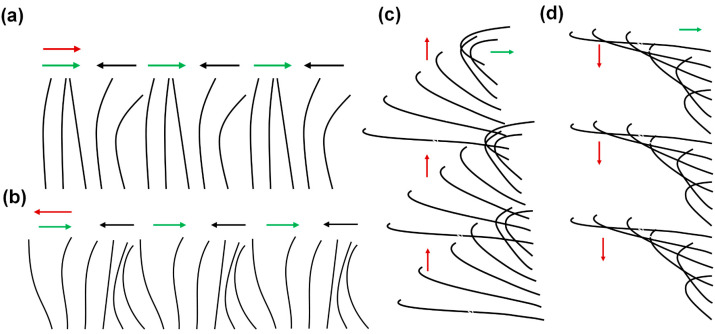
Schematics of the four types of metachronal motion exhibited by biological cilia. Each diagram represents a row of cilia. The red arrows indicate the direction of the metachronal wave, the green arrows represent the directions of the effective stroke, and the black arrows represent the recovery stroke. (**a**) Symplectic metachronal motion; (**b**) antiplectic metachronal motion; (**c**) dexioplectic metachronal motion; and (**d**) laeoplectic metachronal motion.

**Figure 2 biomimetics-09-00198-f002:**
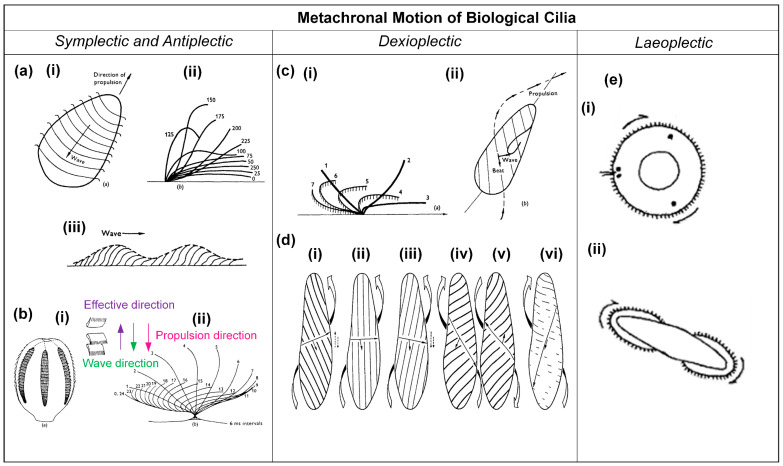
Schematics of examples for the four different types of metachronal motion found in nature. (**a**) (i) Metachronal wave patterns of rows of cilia on the protozoon *Opalina*, exhibiting symplectic metachrony; (ii) the beating of an individual cilium on *Opalina*, with the effective stroke to the right; (iii) the envelope over the metachronal wave of *Opalina*, where the arrow indicates the direction of the metachronal wave, coinciding with the effective stroke direction. Reproduced from ref. [5] with permission from John Wiley and Sons. (**b**) (i) Diagram showing the arrangement of combe-plates of *Pleurobrachia*, as well as the metachronal wave direction and the propulsion direction; (ii) the beat cycle of Pleurobrachia, with the effective stroke to the right; the wave propagation direction is against this, i.e., the metachrony is antiplectic. Reproduced from ref. [5] with permission from John Wiley and Sons. (**c**) Features of *Paramecium*. (i) Schematic of the cilia beating shape during one cycle, with the effective stroke to the right; (ii) schematic of the metachronal wave, which is dexioplextic, and the direction of propulsion. Reproduced from ref. [5] with permission from Wiley. (**d**) Metachronism and locomotion of *Paramecium* for different viscosities of the medium; (i) normal conditions, in medium with a viscosity of 1 cP, showing dexioplectic metachrony; (ii) in medium with a viscosity of 2.6 cP; (iii) in medium with a viscosity of 5.6 cP; (iv) in medium with a viscosity of 40 cP, forward swimming; (v) in medium with a viscosity of 40 cP, backward swimming; (vi) in medium with a viscosity of 135 cP, symplectic metachrony. Reproduced from ref. [4] with permission from The Company of Biologists. (**e**) (i) Larva of Bugula, viewed apically; (ii) Cyphonautes larva of bryozoan, viewed apically. The metachronal wave is indicated by the arrows; the effective stroke of the individual cilia is pointing into the plane of view; hence, the metachrony is laeoplectic. Reproduced from ref. [5] with permission from John Wiley and Sons.

**Figure 3 biomimetics-09-00198-f003:**
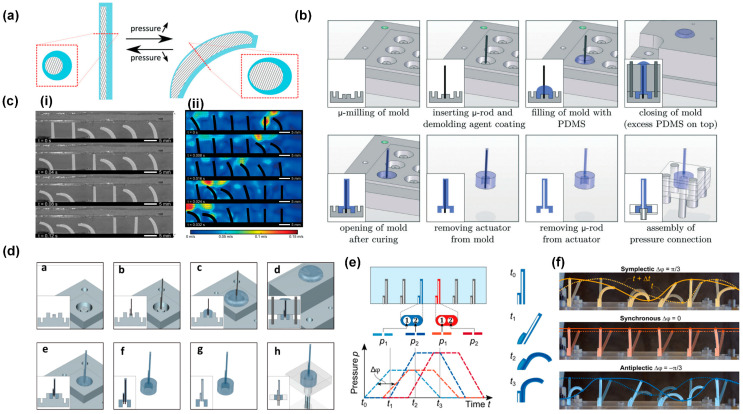
Metachronal motion by pneumatic cilia. (**a**) Schematic of the principle of a flexible bending actuator that consists of an asymmetric void (hatched) surrounded by a highly flexible material; the cross-section view is shown in the zoom-in figure. (**b**) The fabrication process of the pneumatic artificial cilia. (**c**) (i) Experimental results showing an antiplectic metachronal wave with a phase difference between adjacent cilia of 45 degrees; (ii) a symplectic metachronic wave with a phase difference between cilia of 45 degrees. Reproduced from ref. [26] with permission from the Royal Society of Chemistry. (**d**) Fabrication process of a pneumatic actuator with more degrees of freedom, enabling an asymmetric cilia stroke. (**e**) Pressure input functions: each cilium is actuated with two trapezoidal waves. A metachronal wave is applied by shifting the trapezoidal waves of the neighboring cilium by a constant phase angle. (**f**) An array of six artificial pneumatic cilia independently actuated by 12 fluid pressure inputs. Symplectic and antiplectic waves, as well as synchronous motion, can be applied to the array. Reproduced from ref. [21] with permission from the American Association for the Advancement of Science.

**Figure 4 biomimetics-09-00198-f004:**
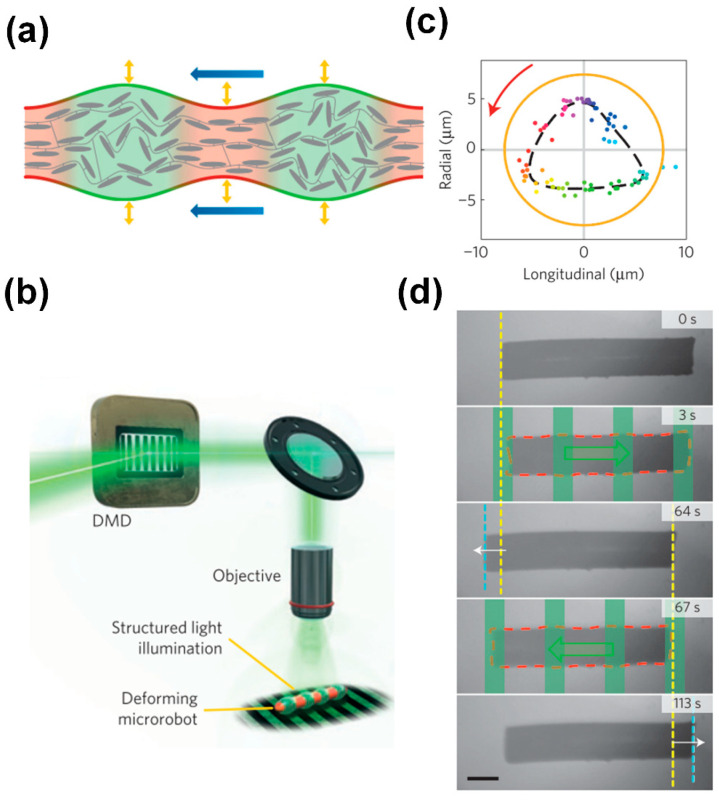
Light-driven metachronal motion. (**a**) Local illumination of a liquid crystal polymer that incorporates azobenzene can induce local contraction and expansion. (**b**) A Digital Micromirror Device (DMD) can generate structured light fields that can induce complex wavelike motion. (**c**) By controlling illumination conditions, the relative amplitudes of longitudinal and axial deformations can be manipulated. (**d**) Back and forth swimming of a cylindrical microrobot propelled by traveling-wave deformations (red dashed line: deformed profile). The green overlays and arrows represent the periodic light pattern and its traveling direction, respectively. Yellow and cyan dashed lines represent the initial and final positions of the leading edge of the robot, respectively. Reproduced from ref. [29] with permission from Nature Portfolio.

**Figure 5 biomimetics-09-00198-f005:**
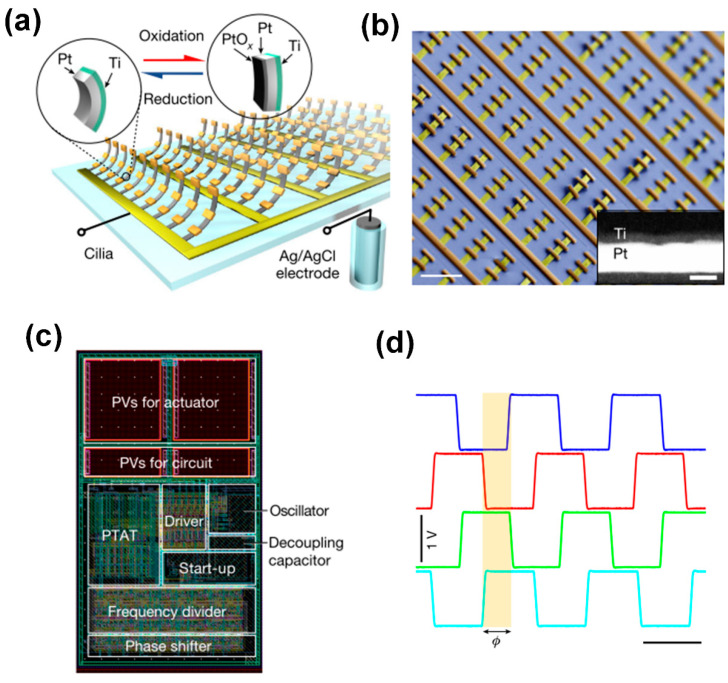
Metachronal motion of electrically driven artificial cilia. (**a**) Artificial cilia array based on surface electrochemical actuators; each cilium consists of a thin platinum strip capped on one side by a titanium film. (**b**) SEM image of a released artificial cilia array with each row connected by a single busbar; the inset is a STEM image of a cilium cross-section. (**c**) Remote control of the cilia is realized by a CMOS circuit, the layout of which is shown here. (**d**) Four voltage outputs from the CMOS circuit enable metachronal motion of the artificial cilia. Reproduced from ref. [23] with permission from Nature Portfolio.

**Figure 6 biomimetics-09-00198-f006:**
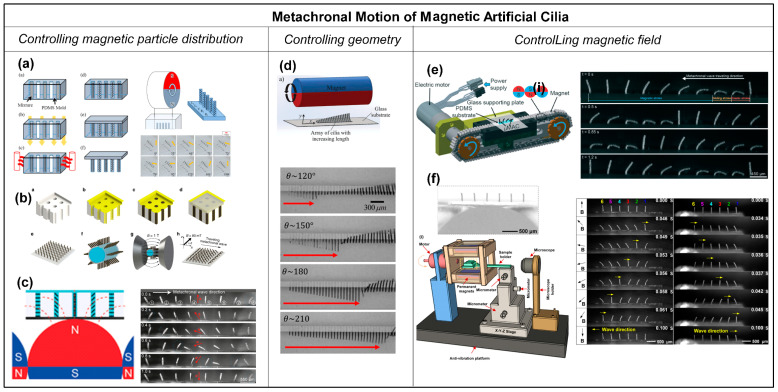
Metachronal motion of magnetic artificial cilia. (**a**) Fabrication process of a cilia array with magnetic particle alignment varying across the array, achieved by a step-by-step filling of the cilia mold and solidifying the material while applying a magnetic field with a changing field angle. Reproduced from ref. [59] with permission from the Japan Society of Applied Physics. (**b**) Fabrication and magnetization process of magnetic artificial cilia carpets with magnetization direction variation across the cilia array by stretching the array (made from NdFeB particles and Ecoflex) around a magnetizing structure. Reproduced from ref. [58] with permission from Nature Portfolio. (**c**) Curing the magnetic cilia array (PDMS and paramagnetic particles) on top of a rod-shaped magnet, leading to varying magnetic particle distribution over the cilia array, and the demonstration of metachronal motion generated by the magnetic cilia array when applying a rotating uniform magnetic field. Reproduced from ref. [57] with permission from the American Chemical Society. (**d**) Magnetic cilia having different lengths actuated with an external rotational magnetic field and the achieved metachronal motion. The arrow indicates the position of the wave front. Reproduced from ref. [50] with permission from the Royal Society of Chemistry. (**e**) Array with identical magnetic artificial cilia, actuated with a translating magnetic belt consisting of rod-shaped magnets arranged with opposite dipoles between adjacent magnets, and the metachronal motion demonstrated for this array. Reproduced from ref. [38] with permission from the Royal Society of Chemistry. (**f**) Array with identical magnetic artificial cilia having a rod-shaped magnetic substructure underneath and the symplectic and antiplectic metachronal motion realized by the method when actuated with a rotational uniform magnetic field. Reproduced from ref. [18] with permission from the National Academy of Sciences.

**Figure 7 biomimetics-09-00198-f007:**
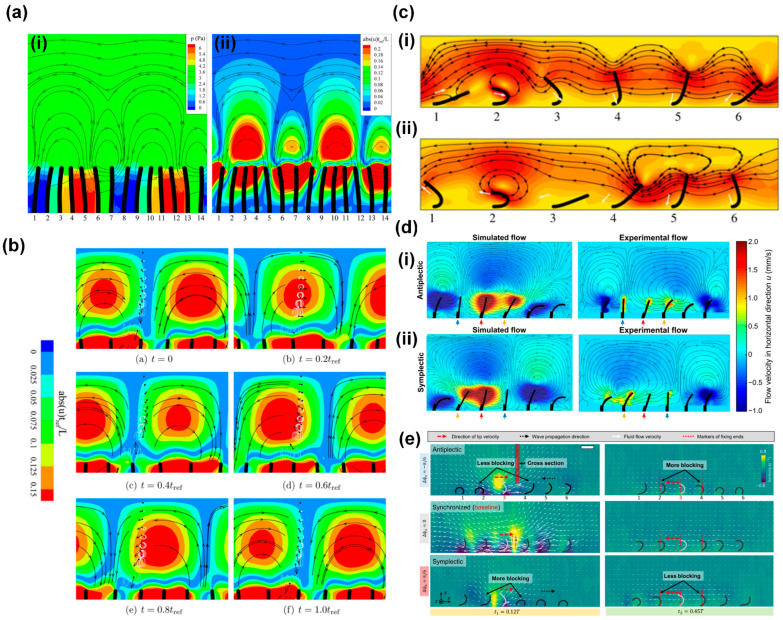
Mechanisms of flow generation by metachronal motion of artificial cilia. (**a**) Numerical simulations of flow generation by symmetrically beating cilia exhibiting metachronal motion, with the metachronal wave traveling to the right, showing (i) pressure contours (red is high pressure and blue is low pressure) and (ii) contours of normalized absolute horizontal velocity; the streamlines represent the direction of velocity. Reproduced from ref. [64] with permission from the American Institute of Physics. (**b**) Numerical simulations of flow generation by symmetrically beating cilia exhibiting metachronal motion, with the metachronal wave traveling to the right, showing the motion of tracer particles with time; the white curves represent the trajectory of particles, and the black dots represent the particles. Reproduced from ref. [64] with permission from the American Institute of Physics. (**c**) Snapshots of antiplectic and symplectic metachronal motion of cilia from numerical simulations; the effective stroke of the non-reciprocally moving cilia is to the left; the contours represent the normalized absolute velocity; the streamlines represent the direction of the velocity; (i) antiplectic metachrony, in which waves travel to the right; and (ii) symplectic metachrony, in which waves travel to the left. Reproduced from ref. [63] with permission from Cambridge University Press. (**d**) Simulation and experimental results for pneumatically actuated artificial cilia for both (i) antiplectic and (ii) symplectic metachrony; the color represents the absolute value of generated flow velocity, and the streamlines indicate the direction of flow. Reproduced from ref. [21] with permission from the American Association for the Advancement of Science. (**e**) Experimentally observed fluid flow distribution generated by a magnetic artificial cilia array with antiplectic, synchronized, and symplectic metachrony. The cilia in the array undergoing antiplectic metachrony have less blocked local fluid flow from their neighbors during the power stroke than in the synchronous case and more blocked local fluid flow during the recovery stroke; this is the other way around for symplectic metachrony. Reproduced from ref. [35] with permission from the American Association for the Advancement of Science.

**Figure 9 biomimetics-09-00198-f009:**
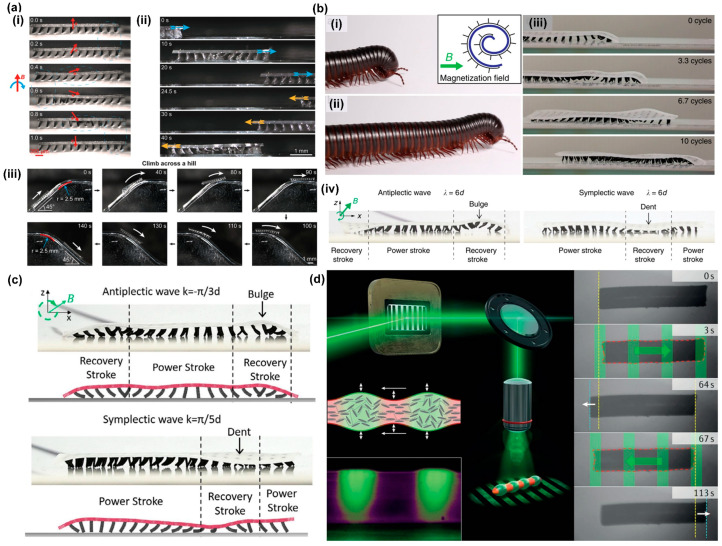
Microrobot locomotion realized by the metachronal motion of artificial cilia. (**a**) (i) A walking metachronal microrobot with metachronal magnetic artificial cilia as legs under a rotational uniform magnetic field in air during one beating cycle; (ii) demonstration of the bi-directional walking capability of the metachronal robot, achieved by reversing the rotating direction of the external magnetic field; (iii) demonstration of the climbing ability of the microrobot across a steep hill. Reproduced from ref. [57] with permission from the American Chemical Society (**b**) (i) and (ii) a crawling giant African millipede with legs moving in traveling metachronal waves; (iii) crawling magnetic soft microrobots inspired by the giant African millipede; (iv) a curved body of antiplectic and symplectic wave soft robots; the body of the antiplectic soft microrobot bulges at the location of the recovery stroke, helping the legs to freely move; the body of the symplectic soft microrobot dents and obstructs the recovery stroke, which slows down the robotic locomotion. Reproduced from ref. [58] with permission from Nature Portfolio. (**c**) Robot body deformation for antiplectic and symplectic wave soft robots, compared between experiments (photos) and simulations (drawings). Reproduced from ref. [84] with permission from John Wiley & Sons. (**d**) A soft microrobot based on a photoresponsive liquid–crystal elastomer swims by traveling-wave deformations, mimicking metachronal waves in ciliates. Reproduced from ref. [29] with permission from Nature Portfolio.

## Data Availability

There were no new data created in this article.

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
