# Peer review of "Metachronal Motion of Biological and Artificial Cilia"

_biomimetics, 2024, doi:10.3390/biomimetics9040198_

Round 1
Reviewer 1 Report
Comments and Suggestions for Authors
The proposed review considers arrays of cilia and their motion, where beating of each individual cilium is coordinated leading to certain types of wave-like motion. Studies on biological cilia arrays, manufacturing and simulation of artificial cilia arrays are presented. Possible applications are discussed. In the reviewer's opinion the work may be published after a minor revision.
1) Line 195
When concerning the transport of the ovum, the reference to [50] is given, but [50] proposes some [magnetic] design of a cilia array and mentions the transport of the ovum in only one sentence, namely,
“Cilia serve a wide variety of biological functions [ref1]. For example, the beating of cilia in the fallopian tubes moves the ovum from the ovary to the uterus [ref2].”, where [ref2] is the biological study
(2) Lyons, R.; Saridogan, E.; Djahanbakhch, O. The reproductive significance of human Fallopian tube cilia. Human reproduction update 2006, 12 (4), 363-372. doi: 10.1093/humupd/dml012
As far as the corresponding section in your paper is devoted to the biological cilia, maybe it would be more appropriate to site this biological paper or to site both [50] and [ref2].
2) The review mentions papers on biological and artificial cilia arrays (experimental and with simulation). The review would benefit if some analytical studies of the mentioned or possibly simplified systems were added (if any are known to the authors).
Some minor inaccuracies and typos:
3) Line 53
It seems that the coma is missed after “mucociliary clearance [1,9,12–15]”
4) Line 234
The acronym PDMS should be defined earlier (when firstly mentioned, Line 219)
5) Line 244
Full stop in “[21.]”
6) Line 545, “motion”
Must be misspell of “notion”.
6) Line 597, “belt a shown”
Presumably, "a" should be deleted or changed to "as".
7) Line 750
The acronym LCE is not defined. (Should be defined earlier in Line 748).
8) Line 813
Misspell “physiolocial”
9) Please, check the correctness of writing for items [8], [19], and [55] in References section.
Author Response
We thank the reviewers for their thoughtful and encouraging remarks. We appreciate their critical and useful comments that have helped us to improve our manuscript. In the following text, we specifically address the points of the reviewers with our response in green. The corresponding changes are highlighted in red in the revised manuscript.

Reviewer 2 Report
Comments and Suggestions for Authors
This article meticulously explores the metachronal motion exhibited by both biological and artificial cilia, highlighting the pivotal roles these hair-like structures play in fluid flow propulsion and diverse functions across various organisms. The comprehensive examination encompasses different types of artificial cilia, employing pneumatic, photonic, electric, and magnetic mechanisms. The applications of metachronal cilia motion, ranging from flow generation to particle transportation and microrobot locomotion, are thoroughly discussed. The document adeptly underscores various fabrication techniques for artificial cilia and delves into their potential applications in controlled particle transport and microrobotics. The article stands as a commendable review paper with a strong grasp of cilia development. Despite its well-organized structure and rich content, the article acknowledges the need for improvement in key areas before official publication.
1.To enhance the document, the author is advised to improve clarity and structure by ensuring a well-defined and logically organized approach in each section, including the introduction, methods, results, and conclusion.
2.The inclusion of more visual aids such as figures, charts, and diagrams is recommended to augment reader understanding and engagement. The author should expand on the comparative analysis of different types of artificial cilia, providing a detailed discussion on their respective advantages, disadvantages, and applications for informed comparisons.
3.Addressing more potential challenges and limitations associated with artificial cilia use would provide a more comprehensive perspective on practical implications and potential obstacles faced by researchers.
Author Response

(The authors gave the same response as above.)

Reviewer 3 Report
Comments and Suggestions for Authors
Interesting paper,
perform the improvements:
- The introduction could provide more background on the biological significance and functions of metachronal cilia motion to further establish its importance. cite doi:10.1007/s00405-022-07267-0
- The methods section is unclear - consider including a flow chart or schematic overview of the literature review process and selection criteria.
- The results lack quantitative analysis - inclusion of tables/graphs summarizing key findings from the literature would greatly enhance this section.
- More critical discussion of limitations of current artificial cilia technologies and challenges to mimicking biological metachrony is needed. cite PMCID: PMC9063641.
- Discuss different diseases with cilia disorders as CRS and Kartagener Syndrome. cite doi:10.1007/s00405-023-08184-6 and PMID:19182755.
- Conclusions would benefit from a more balanced perspective - call for advanced materials, improved fabrication etc. before claiming potential for diverse applications.
To strengthen the manuscript, I recommend:
- Expanding the introduction to provide more biological context.
- Adding a clear description of the literature review methodology.
- Incorporating visual summaries of the key results/findings.
- Enhancing the discussion of limitations and challenges facing the field.
- Softening the conclusions to reflect the preliminary stage of artificial cilia technologies.
any concerns
Author Response

(The authors gave the same response as above.)
